# 5-Hydroxymethylcytosine in Cell-Free DNA Predicts Immunotherapy Response in Lung Cancer

**DOI:** 10.3390/cells13080715

**Published:** 2024-04-19

**Authors:** Jianming Shao, Yitian Xu, Randall J. Olsen, Saro Kasparian, Kai Sun, Sunil Mathur, Jun Zhang, Chuan He, Shu-Hsia Chen, Eric H. Bernicker, Zejuan Li

**Affiliations:** 1Department of Pathology and Genomic Medicine, Houston Methodist Hospital, Houston, TX 77030, USArjolsen@houstonmethodist.org (R.J.O.); 2Houston Methodist Research Institute, Houston, TX 77030, USAsmathur2@houstonmethodist.org (S.M.); schen3@houstonmethodist.org (S.-H.C.); 3Weill Cornell Medical College, New York, NY 10065, USA; 4Neal Cancer Center, Houston Methodist Hospital, Houston, TX 77030, USAehbernick@gmail.com (E.H.B.); 5Department of Medical Oncology, City of Hope Comprehensive Cancer Center, Duarte, CA 91010, USA; 6Department of Chemistry, Department of Biochemistry and Molecular Biology, Institute for Biophysical Dynamics, The University of Chicago, Chicago, IL 60637, USA; 7Howard Hughes Medical Institute, The University of Chicago, Chicago, IL 60637, USA

**Keywords:** cell-free DNA, 5-hydroxymethylcytosine, immune checkpoint inhibitor, lung cancer, predictive, biomarker

## Abstract

Immune checkpoint inhibitors (ICIs) drastically improve therapeutic outcomes for lung cancer, but accurately predicting individual patient responses to ICIs remains a challenge. We performed the genome-wide profiling of 5-hydroxymethylcytosine (5hmC) in 85 plasma cell-free DNA (cfDNA) samples from lung cancer patients and developed a 5hmC signature that was significantly associated with progression-free survival (PFS). We built a 5hmC predictive model to quantify the 5hmC level and validated the model in the validation, test, and control sets. Low weighted predictive scores (wp-scores) were significantly associated with a longer PFS compared to high wp-scores in the validation [median 7.6 versus 1.8 months; *p* = 0.0012; hazard ratio (HR) 0.12; 95% confidence interval (CI), 0.03–0.54] and test (median 14.9 versus 3.3 months; *p* = 0.00074; HR 0.10; 95% CI, 0.02–0.50) sets. Objective response rates in patients with a low or high wp-score were 75.0% (95% CI, 42.8–94.5%) versus 0.0% (95% CI, 0.0–60.2%) in the validation set (*p* = 0.019) and 80.0% (95% CI, 44.4–97.5%) versus 0.0% (95% CI, 0.0–36.9%) in the test set (*p* = 0.0011). The wp-scores were also significantly associated with PFS in patients receiving single-agent ICI treatment (*p* < 0.05). In addition, the 5hmC predictive signature demonstrated superior predictive capability to tumor programmed death-ligand 1 and specificity to ICI treatment response prediction. Moreover, we identified novel 5hmC-associated genes and signaling pathways integral to ICI treatment response in lung cancer. This study provides proof-of-concept evidence that the cfDNA 5hmC signature is a robust biomarker for predicting ICI treatment response in lung cancer.

## 1. Background

Lung cancer is the third most common cancer and the leading cause of cancer-related death in the United States [1]. Despite significant advances in screening, diagnosis, and treatment, the five-year survival rate for lung cancer patients is only 22% [1]. Immune checkpoint inhibitors (ICIs) targeting programmed cell death 1 (PD-1), its ligand (PD-L1), and cytotoxic T-lymphocyte-associated protein 4 (CTLA-4) have emerged as a promising treatment option for lung cancer patients and can dramatically improve clinical outcomes [2,3,4,5]. Indeed, current therapeutic protocols for advanced non-small cell lung cancer (NSCLC) are transitioning from traditional chemotherapy toward therapies based on tyrosine kinase inhibitors or ICIs [6].

However, a remaining clinical challenge lies in identifying patients who will positively respond to ICI therapy. Current U.S. Food and Drug Administration-approved biomarkers, such as tumor PD-L1 expression, do not adequately identify patients suitable for ICI therapy [3,4,7,8,9]. Consequently, nearly all metastatic NSCLC patients receive ICIs in conjunction with standard chemotherapy in first-line settings, with the exception of those harboring targetable driver mutations [5]. However, many ICI-treated patients reap no benefits and up to 87% suffer various immune-related adverse events, some of which are potentially fatal [5,7,10]. Therefore, more sensitive predictive biomarkers that can identify patients likely to benefit from ICI treatment are urgently needed to prevent avoidable side effects and mitigate economic burden.

Global loss of 5-hydroxymethylcytosine (5hmC) occurs in lung and many other cancers [11,12,13,14,15,16,17,18]. 5hmC, an oxidation product converted from 5-methylcytosine (5mC) by ten-eleven translocation proteins (TET), is a dynamic marker that closely correlates with gene expression [11]. Aberrant 5hmC levels are associated with the onset, progression, and metastasis of cancer [11]. A low 5hmC level in tumor tissue is significantly associated with poor prognosis in lung cancer [19]. 5hmC levels change dynamically during T cell development and differentiation and are associated with the ability of tumors to evade immune detection and induce T cell exhaustion [20,21,22,23]. 5hmC also affects key immune genes during T lymphocyte activation and differentiation following antigen presentation [20,21,24,25]. For example, 5hmC levels are elevated at the PD-1 promoter of non-tolerant T cells but reduced in response to immunotherapy [25]. Alterations of 5hmC are implicated in both innate and acquired resistance to ICIs via both tumor-intrinsic and extrinsic pathways [21]. Moreover, mutations in *TET1* and *TET2* are associated with improved overall survival (OS) in ICI-treated lung cancer patients [26,27]. Most recently, studies revealed that ICI treatment induced distinct changes in plasma cell-free DNA (cfDNA) 5hmC profiles in lung cancer patients [28]. These observations suggest that 5hmC is a potential marker for predicting the ICI treatment response.

Using a highly sensitive nano-hmC-Seal method [29], we and others have demonstrated that 5hmC signatures in plasma cfDNA are highly sensitive for the early detection and prognosis of diverse cancers, including lung cancer [30,31,32,33,34,35,36,37,38,39,40,41,42]. However, the predictive value of cfDNA 5hmC in lung cancer immunotherapy response prediction remains unexplored. To evaluate the utility of cfDNA 5hmC for ICI treatment response prediction in lung cancer patients, we profiled the genome-wide distribution of 5hmC in 85 plasma cfDNA samples from lung cancer patients using nano-hmC-Seal method alongside next-generation sequencing (nano-hmC-Seal-Seq). We discovered a cfDNA 5hmC signature and developed and validated a 5hmC predictive model for predicting immunotherapy response in lung cancer. The 5hmC signature was more accurate than tumor PD-L1 for ICI treatment response prediction. Furthermore, we identified previously unrecognized genes and signaling pathways with aberrant 5hmC levels, providing critical insight into the molecular underpinnings of ICI treatment response in lung cancer.

## 2. Methods

### 2.1. Patients and Sample Collection

A total of 85 samples of whole blood (3–5 mL) were obtained from 83 adult patients with lung cancer (stage III or IV) at Houston Methodist Hospital between 2015 and 2022 (Appendix A). Of these patients, 18 were enrolled in a STOMP clinical trial, which was designed to evaluate the treatment effect of the oncolytic virus and pembrolizumab; these patients received the oncolytic virus 15–47 days before receiving pembrolizumab monotherapy. Another group of 40 patients received either ICI monotherapy [pembrolizumab (n = 10), nivolumab (n = 2), durvalumab (n = 2), or atezolizumab (n = 1)] or ICI in combination with chemotherapy (n = 25). Within this group, two patients joined the STOMP trial after starting their initial ICI treatment and were analyzed separately based on their respective treatment regimens. The ICIs included pembrolizumab, nivolumab, atezolizumab, and durvalumab. In a third group, 27 patients were treated with chemotherapy, including carboplatin, pemetrexed, paclitaxel, and/or etoposide, and/or targeted therapy with tyrosine kinase inhibitors, including erlotinib, osimertinib, alectinib, or cabozantinib, but did not receive ICIs. Tumor-nodes-metastasis (TNM) staging was determined according to guidelines in the 8th edition of the TNM Classification specific to lung cancer [43]. Tumor PD-L1 expression was assessed in 74 patients via immunohistochemical assay using anti-PD-L1 (SP142).

For STOMP clinical trial patients, 13 blood samples were collected on the same day prior to pembrolizumab treatment, and 5 samples were collected between 1 and 29 days after beginning pembrolizumab treatment. For other patients receiving ICIs, blood samples were collected before the initiation of ICI treatment (34 samples; 0.0–2.6 months, median 0.68 months) or after initial therapy (6 samples; 0.80–7.4 months, median 2.6 months). For the 27 patients in the non-ICI group, 18 blood samples were collected before the initiation of therapy (0.03–1.5 months, median 0.60 months), four were collected after therapy (0.16–6.1 months, median 1.8 months), and five were from patients who did not receive systemic treatment.

### 2.2. Study Design

We divided 85 plasma cfDNA samples from lung cancer patients into four sets—training, validation, test, and control (Appendix A; Appendix A). Samples (n = 58) from lung cancer patients treated with ICI-based therapies were used for the 5hmC signature discovery and validation. Forty samples were collected from lung cancer patients who received ICI treatment and were then randomly assigned into a training (n = 24) or a validation set (n = 16). Eighteen patients enrolled in the STOMP clinical trial for oncolytic virus plus pembrolizumab treatment were used as a test set. Twenty-seven late-stage lung cancer patients who did not receive ICI treatment were used as a control set. We performed the genome-wide profiling of 5hmC in the plasma cfDNA samples from all patients. We first correlated 5hmC distribution with progression-free survival (PFS) to identify a 5hmC predictive signature in the training set. This signature was then blindly validated in the validation, test, and control sets. We also compared the 5hmC signature with the PD-L1 tumor proportion score (TPS) for ICI treatment response prediction in patients with PD-L1 analysis. To delineate the role of 5hmC in immunotherapy, we analyzed genes and canonical signaling pathways in which aberrant 5hmC levels were significantly associated with ICI treatment in lung cancer patients. The primary endpoint was PFS or death from any cause. Secondary endpoints were OS and safety. All patients were followed up for a minimum of six months.

### 2.3. Treatment Response Assessments

PFS was defined as the time from the date of patient registration until the occurrence of tumor progression, a switch to another therapy, or death from any cause, whichever occurred first. If patients were lost to follow-up, they were censored at the last known contact. OS was defined as the time between the date of patient registration to the date of death from any cause, or the last follow-up, whichever occurred first. An objective response was evaluated based on the Response Evaluation Criteria in Solid Tumors (RECIST) guidelines, version 1.1 [44]. Patient treatment responses were evaluated after blood collection time points. Patients showing complete or partial responses were classified as responders, while those exhibiting progressive or stable disease were considered non-responders. Objective response rate (ORR) was defined as the percentage of patients who achieved a complete or partial response to treatment within six months of initial blood collection.

### 2.4. Sample Processing and 5hmC Sequencing Analysis

cfDNA was isolated from approximately ~1 mL of plasma and the 5hmC library was constructed as previously described [39]. The enriched library was then sequenced in a NovaSeq 6000 instrument (Illumina, San Diego, CA, USA). Following sequencing, data processing was performed as described previously with minor modifications [39]. To quantify high-quality reads within gene bodies (RefSeq), the featureCounts, version 2.0.0, tool was used. Genes with low read counts (<3 per million) in over 50% of samples were excluded from subsequent analysis.

### 2.5. Establishing 5hmC Predictive Signatures

A 5hmC predictive signature was developed as previously described [38]. Briefly, we randomly split non-STOMP samples into a training set and a validation set at a ratio of 6–4. In the training set, we correlated 5hmC levels with PFS using a univariate Cox proportional-hazards regression model and identified genes significantly associated with PFS (*p* < 0.05). Feature selection was performed using elastic net regularization (α range: 0.55–0.95 with 0.1 increments) applied to a multivariate Cox proportional-hazards model via the glmnet package, version 4.0. The selection process was repeated 100 times. Genes appearing in at least 95% of iterations were identified as signature genes and used to develop the final 5hmC model. We calculated a weighted-predictive score (wp-score) based on the model for each sample. Cut-off scores were determined in the training sets using the surv_cutpoint function from the survminer R package (https://github.com/kassambara/survminer; accessed on 26 July 2022).

### 2.6. Power and Statistical Analyses

In a one-way ANOVA study that allows for unequal group variances, a sample of 58 subjects, divided among three groups (training, validation, and test), achieves a power of 95%. This power assumes the data will be analyzed with Welch’s test with a significance level of 0.05. The group subject counts are 24 patients (training), 16 patients (validations), and 18 patients (test). The group means under the null hypothesis are assumed to be equal. Based on our previous experience, the group means under the alternative hypothesis are 0, 1, 2. The group standard deviations are assumed as 1, 1.2, 2.2. The standard deviation of the standardized means under the alternative hypothesis is 0.543. Kaplan–Meier estimates were used to approximate the PFS and OS and the log-rank test was used to calculate differences between groups. Univariate and multivariate Cox proportional-hazards regression models were used to calculate hazard ratios and identify independent variables associated with PFS and OS, respectively. The ‘timeROC’ R package, version 0.4, was used to assess the performance of the 5hmC prognostic model [45]. Forest plots were generated using forestplot (https://cran.r-project.org/web/packages/forestplot/index.html, accessed on 21 August 2023). All plotting and statistical tests were performed using R language version 4.1.1. Comparisons of the accuracy of the prediction markers were evaluated using a two-tailed Fisher’s exact test. Comparisons of wp-scores between groups were analyzed using the Wilcoxon Rank Sum test. Gene enrichment analyses were performed using Ingenuity Pathway Analysis (IPA). A *p* value < 0.05 was considered statistically significant.

## 3. Results

### 3.1. A 5hmC Predictive Signature Is Associated with PFS in ICI-Treated Patients

In the training set, we correlated 5hmC levels with PFS and identified 104 genes significantly associated with PFS (*p* < 0.05; Appendix A). We discovered a 16-gene signature through feature selection and built a 5hmC predictive model based on these signature genes (Appendix A). We then calculated a wp-score that represented the 5hmC level of the signature genes for each sample (Appendix A). We defined a cut-off wp-score of 175.0 to maximize the sensitivity for detecting patients with ICI treatment response. Low wp-scores were significantly associated with a longer PFS compared to high wp-scores [median 12.3 versus 3.0 months; *p* = 3.5 × 10^−6^; HR: 4.6 × 10^−10^; 95% confidence interval (CI), 0–infinite; Figure 1A]. The estimated proportion of patients who were alive without disease progression at six months was 100.0% among those with a low wp-score and 27.3% for those with a high wp-score (Figure 1A).

This result was confirmed in independent patient cohorts. In the validation set, the median PFS was 7.6 months in patients with low wp-scores versus 1.8 months for those with high wp-scores (*p* = 0.0012; HR 0.12; 95% CI, 0.03–0.54; Figure 1B). At six months, the estimated proportion of patients alive without disease progression was 75.0% for those with low wp-scores, while no patients with a high wp-score demonstrated a PFS beyond this duration (Figure 1B). In the test set, the median PFS was 14.9 months in individuals with low wp-scores versus 3.3 months in those with high wp-scores (*p* = 0.00074; HR 0.10; 95% CI, 0.02–0.50; Figure 1C). At six months, the estimated proportion of patients who were alive without disease progression was 80.0% for those with low wp-scores versus 0.0% for those with high wp-scores (Figure 1C). Multivariate Cox regression analysis demonstrated that the 5hmC predictive signature was significantly correlated with PFS, independently of age, sex, race, and PD-L1 TPS (Appendix A).

### 3.2. The 5hmC Predictive Signature Is Associated with Objective Response Rate

ORR was 100.0% (95% CI, 71.5–100.0%) in patients with low wp-scores and 20.0% (95% CI, 2.5–55.6%) in individuals with high wp-scores in the training set (*p* = 0.00022; Table 1). Similarly, in the validation set, the ORR was 75.0% (95% CI, 42.8–94.5%) in individuals with low wp-scores and 0.0% (95% CI, 0.0–60.2%) in those with high wp-scores (*p* = 0.019; Table 1). In the test set, the ORR was 80.0% (95% CI, 44.4–97.5%) in patients with low wp-scores and 0.0% (95% CI, 0.0–36.9%) in patients with high wp-scores (*p* = 0.0011; Table 1). The wp-score achieved an area under the curve (AUC) of 100.0% (95% CI, 100.0–100.0%) in the training set, 96.8% (95% CI, 89.4–100.0%) in the validation set, and 88.8% (95% CI, 70.9–100.0%) in the test set (Figure 2A). Wp-scores were significantly lower in patients who responded to the ICI treatment compared to non-responders in the training (*p* = 0.0073), validation (*p* = 0.00070), and test (*p* = 0.0044) sets (Figure 2B).

### 3.3. The 5hmC Predictive Signature Is Associated with Overall Survival in Patients Receiving ICIs

To assess whether the 5hmC predictive signature was associated with OS, we correlated wp-scores with OS in ICI-treated patients. Low wp-scores were significantly associated with a longer OS in the training set (median 23.2 versus 7.4 months; *p* = 0.00017; HR 0.046, 95% CI, 0.0054–0.39; Appendix A). However, the association between the wp-scores and OS in patients who received ICI treatment was not significant in the validation set (median 10.5 versus 9.2 months; *p* =0.98; HR 1.0; 95% CI, 0.11–9.30; Appendix A) and borderline significant in the test set (median 30.2 versus 16.4 months; *p* = 0.064; HR 0.35; 95% CI, 0.11–1.10; Appendix A). The estimated proportions of patients alive at 12 months with a low or high wp-scores were 100.0% or 45.4% in the training set, 59.4%, or 50.0% in the validation set, and 80.0% or 87.5% in the test set (Appendix A), respectively.

### 3.4. The 5hmC Predictive Signature Is Associated with Outcomes in Patients Receiving Single-Agent ICI Treatment

Although the 5hmC signature was developed based on patients receiving ICIs either as monotherapy or in combination with chemotherapy, it was significantly associated with pembrolizumab monotherapy, the most commonly used ICI in lung cancer, in the test set. We therefore evaluated whether the 5hmC signature could predict the treatment response to ICI monotherapy in non-STOMP clinical trial patients (3 from the training set and 12 from the validation set). Low wp-scores were significantly associated with a longer PFS relative to the high wp-scores (median 19.6 versus 2.8 months; *p* = 0.00073; HR: 0.059; 95% CI, 0.0069–0.50; Figure 3A). Although not significant, low wp-scores were associated with a longer OS (median 38.1 versus 10.5 months; *p* =0.12; HR: 0.27; 95% CI, 0.047–1.50; Figure 3B). At six months, the estimated proportion of patients who were alive without disease progression was 87.5% in patients with a low wp-score and 0.0% in patients with a high wp-score (Figure 3A). ORR was 87.5% (95% CI, 47.4–99.7%) in patients with low wp-scores and 0.0% (95% CI, 0.0–41.0%) in patients with high wp-scores (*p* = 0.0014; Table 1).

### 3.5. The 5hmC Predictive Signature Is Superior to Tumor PD-L1 Expression for ICI Response Prediction

Of the non-STOMP patients, 35 had available tumor PD-L1 expression data. A high (≥1%) or a low (<1%) PD-L1 TPS was not significantly associated with PFS (median 6.8 versus 6.0 months; *p* = 0.40; HR 0.70; 95% CI, 0.30–1.60; Appendix A), OS (median 10.1 versus 12.0 months; *p* = 0.38; HR 0.56; 95% CI, 0.15–2.10; Appendix A), or ORR (53.9%, 95% CI, 25.1–80.8% versus 57.9%, 95% CI, 33.5–79.8%; Appendix A) in these patients. Similarly, PD-L1 TPS did not exhibit a significant association with PFS (median 3.8 versus 6.1 months; *p* = 0.81; HR 0.88; 95% CI, 0.30–2.50; Appendix A), OS (median 16.8 versus 19.7 months; *p* = 0.94; HR 0.96; 95% CI, 0.33–2.80; Appendix A), or ORR (37.5%, 95% CI, 8.5–75.5% versus 50.0%, 95% CI, 18.7–81.3%; Appendix A) in STOMP clinical trial patients. Compared to PD-L1 prediction, the cfDNA 5hmC signature demonstrated a significantly higher accuracy for the ICI treatment response prediction. The accuracy was 84.4% (67.2–94.7%) versus 46.9% (95% CI, 29.1–65.3%; *p* = 0.0033) in non-STOMP patients and 88.9% (95% CI, 65.3–98.6%) versus 44.4% (95% CI, 21.5–69.2%; *p* = 0.012) in STOMP clinical trial patients (Appendix A).

### 3.6. The 5hmC Predictive Signature Is Specific to ICI Treatment Prediction

To determine whether the 5hmC predictive signature was specific to the response to ICI treatment, we calculated the wp-scores using the 5hmC predictive signature in 27 late-stage lung cancer patients who did not receive ICI treatment. The wp-scores were not significantly associated with PFS (*p* = 0.66; HR 0.81; 95% CI, 0.32–2.00; Appendix A) or OS (*p* = 0.74; HR 0.83; 95% CI, 0.28–2.50; Appendix A) in these patients. This finding suggests that the plasma cfDNA 5hmC predictive signature is not suitable for predicting the response to treatments other than ICIs in lung cancer patients. This result also highlights the specificity of the 5hmC signature in predicting responses to ICI treatments in lung cancer, underscoring its potential as a biomarker specific to this therapeutic context.

### 3.7. Genes and Pathways Associated with ICI Treatment Response

To delineate the molecular mechanisms underlying the ICI treatment response, we analyzed 104 genes for which aberrant 5hmC levels were significantly associated with PFS among ICI-treated lung cancer patients (Appendix A). These genes were significantly enriched in eight canonical signaling pathways, including nucleotide excision repair and enhanced pathway, xenobiotic metabolism general signaling pathway, and sucrose degradation V pathways (*p* < 0.05; Figure 4A; Appendix A). Genes associated with cell immune functions were enriched in these pathways, such as *COPS5*, *GSTM5*, and *TPI1* (Figure 4B; Appendix A), indicating the critical role of 5hmC modifications in immune response in lung cancer treatment.

## 4. Discussion

Our study offers proof-of-concept evidence that 5hmC levels in plasma cfDNA may have a predictive value for ICI treatment response in lung cancer. Because 5hmC levels change dynamically during tumorigenesis and tumor immune responses [20,21,24,25,26,27], 5hmC has the potential to predict the treatment response to ICIs in cancer. As 5hmC closely correlates with gene expression [11], it may also serve as a reliable marker for changes in tumor gene expression in response to ICI treatment. Previous studies indicate that cfDNA 5hmC has a high sensitivity for cancer detection and prognosis prediction [30,31,32,33,34,35,36,37,38,39,40,41,42]. We further add to this body of evidence by demonstrating that cfDNA 5hmC can predict the therapeutic response in lung cancer patients receiving ICI therapy. The accurate prediction of ICI treatment response can help avoid unnecessary side effects and financial burdens for patients.

Our study underscores the significant predictive value of cfDNA 5hmC for patients receiving ICI as a single-agent therapy, which goes beyond its potential for predicting responses to various ICI treatment regimens, as demonstrated in both non-STOMP and STOMP clinical trial patients. Previous studies demonstrated that the ORR was 45% in lung cancer patients with PD-L1 TPS ≥ 50% receiving pembrolizumab monotherapy [3,4], which was comparable to the ORR of 61.4% in unselected lung cancer patients receiving a combination of pembrolizumab and chemotherapy [5]. While ICI monotherapy can spare patients from the devastating side effects of chemotherapy, only ~15% of NSCLC patients truly benefit from it, given that 23.2–30.2% of NSCLC patients have a PD-L1 TPS ≥ 50% [3,4,11]. In our study, 10 of 18 (55.6%) patients in the STOMP clinical trial and 8 of 15 (53.3%) patients in the non-STOMP clinical trial (Figure 3) had a low wp-score, suggesting that, using the 5hmC marker, over half of metastatic lung cancer patients may be eligible to receive single-agent ICIs as initial immunotherapy and more patients could benefit from ICI monotherapy than using PD-L1 as a predictive marker.

The cfDNA 5hmC predictive signature outperformed the tumor PD-L1 expression in predicting the ICI treatment response. The ORR in patients with a low wp-score was 75.0–100.0% compared to 37.5–53.9% in patients with a PD-L1 TPS ≥ 1% in our study and 45% in patients with a PD-L1 TPS ≥ 50% receiving pembrolizumab monotherapy in previous reports [3,4]. Several factors could contribute to this discrepancy. First, the PD-L1 expression is typically assessed in tumor samples, which may not reliably reflect dynamic disease progression. Second, due to the heterogeneous nature of tumor tissue, the PD-L1 expression levels could vary within the tumor. Third, various technical issues, such as the choice of PD-L1 antibody clones, cut-off points, and other issues inherent to immunohistochemical analysis could affect the interpretation of PD-L1 expression. In contrast, the plasma cfDNA 5hmC can capture dynamic and spatial heterogeneity in tumors. Moreover, only a minority of patients (23.2–30.2%) overexpress PD-L1 [3,4,11]. Using the 5hmC predictive signature, we found that 54.2–75.0% of patients had a low wp-score (Figure 1), indicating that a larger proportion of patients might benefit from ICI treatment. Thus, this signature could help identify a larger cohort of suitable ICI therapy recipients.

cfDNA 5hmC has several advantages over other potential markers for ICI treatment response prediction. Currently emerging ICI biomarkers span genomic, epigenetic, transcriptomic, and protein levels [7,46]. DNA methylation markers in tumor tissue have been reported for ICI treatment response prediction in lung cancer [47,48]. However, unlike most genome-wide DNA methylation analyses that necessitate bisulfite treatment and a large quantity of DNA, our method protects DNA from chemical damage and can accurately profile 5hmC with as little as 1–2 ng of DNA from plasma [29]. The use of plasma cfDNA represents a safer, simpler, and less invasive alternative to sampling tumor tissue, which is predominantly used for other immunomarkers in lung cancer [7]. Furthermore, while somatic mutations are limited for ICI treatment prediction due to the scarcity of recurrent mutations, 5hmC changes occur in all lung cancer patients [11] and thus can be applied to a broad patient population.

The role of 5hmC in ICI therapy remains poorly understood. Identifying genes and signaling pathways that exhibit significant 5hmC alterations during ICI treatment may reveal novel mechanisms and potential targets for immunotherapy in lung cancer. We found that significant 5hmC changes were present in immune-related genes in response to ICI treatment. For example, *COPS5*, a c-Jun activation domain-binding protein-1, interacts with multiple tumor-related genes, including PD-L1, and involves many cell signaling pathways, such as IL6-Stat3 Signaling, TGF-β Signaling, and NF-κB Signaling [49]. COPS5 stabilizes PD-L1 to promote the tumor immune evasion expression [49]. The overexpression of *COPS5* is reported in many malignancies including lung cancer and associated with a poor prognosis [49,50]. The *GSTM5* expression is associated with macrophage and mast cell activation and positively correlated with prognosis in lung, ovarian, and gastric cancer [51,52,53]. The high expression of *TPI1* (triosephosphate isomerase 1), as a glycolysis-related gene, has been correlated with the low infiltration of lymphocytes in lung cancer and laryngeal squamous cell carcinoma [54,55]. High *TPI1* expression is often inversely related to clinical outcomes in lung adenocarcinoma, laryngeal squamous cell carcinoma, breast cancer, Ewing’s sarcoma, and cholangiocarcinoma [54,55,56,57,58]. The significant association of 5hmC levels of these genes with ICI treatment response indicates a critical role of these genes in immune regulation in lung cancer patients.

Furthermore, we identified genes that exhibited significant changes in 5hmC levels in response to ICI treatment but have not yet been well studied in cell immune response. These include *ACOT1*, *AMDHD1*, *COPS7A*, *ENPP6*, *GDPD1*, *MOCS3*, and *NR1I3*. Some of these genes have been demonstrated to have critical functions other than in one’s immune system. For example, *NR1I3*, the nuclear receptor subfamily 1, group I, member 3, encodes the constitutive androstane receptor (CAR) protein. CAR is a principal regulator of enzymes involved in drug and xenobiotic metabolism [59,60]. The further investigation of these genes in the context of cancer and immunotherapy will contribute to current understanding of cancer immune response. Targeting these genes may be effective in combination with ICI therapy for lung cancer.

Though we found a significant association between low wp-scores and OS in the training set, this was not replicated in the validation and test sets. This discrepancy could be attributed to the small sample size of these two sets and the short observation period for some patients. Increasing the sample size and extending the study duration could potentially improve the predictive capacity of the 5hmC marker for OS. Although our study validated the cfDNA 5hmC predictive signature for PFS in the validation set and clinical trial patients, future multicenter prospective studies involving a larger and more diverse patient population in clinical settings are necessary to further validate this signature. In addition, the 5hmC predictive marker we developed is based on patients who received a variety of ICI treatments. Developing markers with a higher sensitivity for single-agent ICIs, markers specific to single-agent ICIs, or markers that can predict responses to combinations of ICI and chemotherapy could improve the sensitivity of treatment response predictions. Notably, most of the genes with aberrant 5hmC levels identified herein are not well characterized. Further research into their relationship to immune responses will improve our understanding of cancer immunity in ICI treatment.

## 5. Conclusions

In summary, our study provides proof-of-concept evidence that plasma cfDNA 5hmC can accurately predict an ICI treatment response in lung cancer. Similarly to targeted therapy developed for specific driver mutations, the use of novel and innovative markers to guide ICI therapy may improve the precision and effectiveness of treatment while reducing adverse effects. Though further studies are needed to fully understand its role in ICI treatment response, cfDNA 5hmC signatures could offer a powerful, highly sensitive, and minimally invasive tool for guiding the clinical management of lung cancer patients.

## Figures and Tables

**Figure 1 cells-13-00715-f001:**
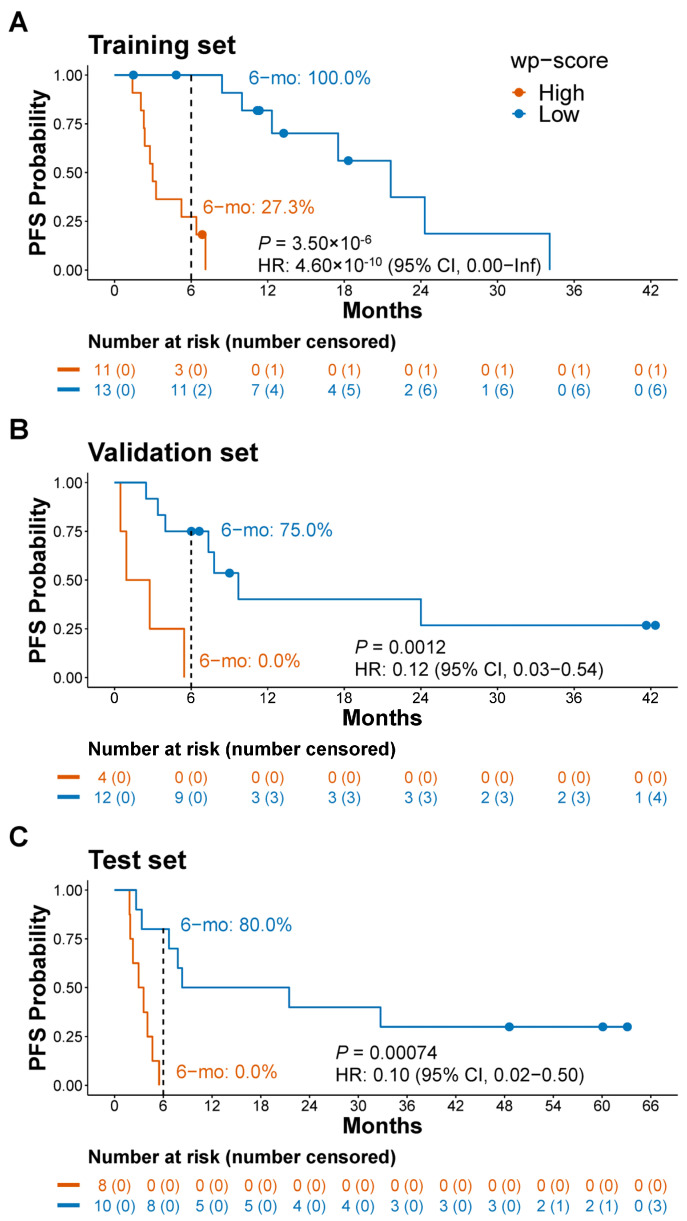
Prediction of progression-free survival by a 5hmC predictive signature in lung cancer patients receiving immune checkpoint inhibitor treatment. (**A**–**C**) Kaplan–Meier analysis of progression-free survival (PFS) based on weighted predictive (wp)-scores in the training set (**A**), the validation set (**B**), and the test set (**C**). 6−mo: estimated PFS in 6 months. Dots on the survival curve indicate that a patient was censored. HR, hazard ratio. CI, confidence interval. Cox proportional-hazards regression model was used to calculate hazard ratio. Log-rank test was used to evaluate the survival difference between groups.

**Figure 2 cells-13-00715-f002:**
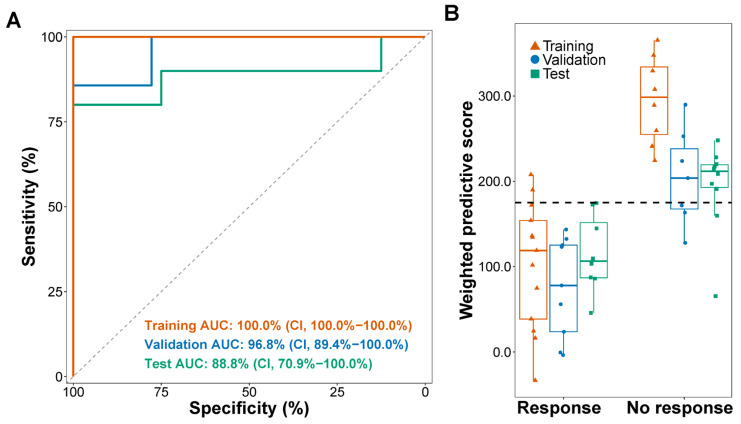
Weighted predictive scores in lung cancer patients receiving immune checkpoint inhibitor treatment. (**A**) Receiver operating characteristics (ROC) analysis of weighted predictive (wp)-scores. AUC, area under the curve. CI, confidence interval. (**B**) Distribution of wp-scores in patients with or without an objective response to immune checkpoint inhibitors in lung cancer patients. Wilcoxon rank sum tests were used to compare the wp-score difference between groups.

**Figure 3 cells-13-00715-f003:**
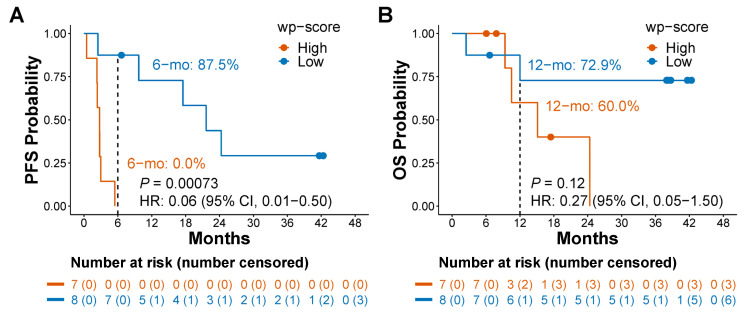
Prediction of survival by a 5hmC predictive signature in lung cancer patients receiving immune checkpoint inhibitor monotherapy. (**A**) Kaplan–Meier analysis of progression-free survival (PFS) based on weighted predictive (wp)-scores. (**B**) Kaplan–Meier analysis of overall survival (OS) based on weighted predictive (wp)-scores. 6−mo: estimated PFS in 6 months. 12−mo: estimated OS in 12 months. Dots on the survival curve indicate that a patient was censored. HR: hazard ratio. CI: confidence interval. Cox proportional-hazards regression model was used to calculate hazard ratio. Log-rank test was used to evaluate the survival difference between groups.

**Figure 4 cells-13-00715-f004:**
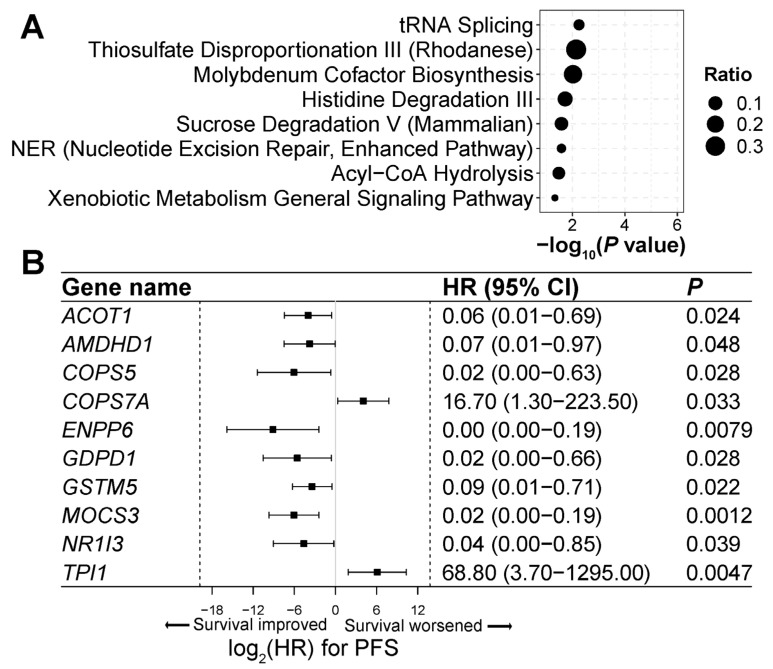
Genes and pathways associated with immune checkpoint inhibitor treatment response in lung cancer. (**A**) Canonical signaling pathways enriched with genes significantly associated with immune check-point inhibitor (ICI) treatment response. Ingenuity pathway analysis (IPA) was used to conduct pathway analysis. Ratio denotes the proportion of genes related to progression-free survival within each pathway to the total number of genes constituting that pathway. (**B**) Hazard ratios for progression-free survival (PFS) in genes significantly enriched in canonical pathways presented by Forest plot. Genes involved in significant canonical pathways are displayed. Univariate Cox proportional-hazards regression model was used to calculate hazard ratio, *p* value < 0.05 is considered as significant.

**Table 1 cells-13-00715-t001:** Objective response to immune checkpoint inhibitors projected by the 5hmC predictive signature.

		Responders (No.)	Non-Responders (No.)	Objective Response Rate (95% CI)
Training	Low wp-score	11	0	100.0% (71.5–100.0%)
	High wp-score	2	8	20.0% (2.5–55.6%)
Validation	Low wp-score	9	3	75.0% (42.8–94.5%)
	High wp-score	0	4	0.0% (0.0–60.2%)
Test	Low wp-score	8	2	80.0% (44.4–97.5%)
	High wp-score	0	8	0.0% (0.0–36.9%)
Single agent	Low wp-score	7	1	87.5% (47.4–99.7%)
	High wp-score	0	7	0.0% (0.0–41.0%)

Patients with treatment response information were included in the training set (n = 21), validation set (n = 16), test set (n = 18), and single-agent patient set (n = 15). Wp, weighted prediction. Responders: patients with complete or partial response to ICI treatment who did not progress within six months after ICI treatment. Non-responders: patients with progression or stable disease or who showed a response but progressed within six months after ICI treatment.

## Data Availability

The raw 5hmC sequencing data supporting the conclusions of this article are available in the National Center for Biotechnology Information Gene Expression Omnibus database, accession number GSE237087.

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
