# Peer review of "5-Hydroxymethylcytosine in Cell-Free DNA Predicts Immunotherapy Response in Lung Cancer"

_cells, 2024, doi:10.3390/cells13080715_

Round 1

Reviewer 1 Report

Comments and Suggestions for Authors

In this study, a 5hmC predictive model was developed to quantify the 5hmC levels in 85 plasma cell-free DNA (cfDNA) samples from lung cancer patients receiving ICI monotherapy. The model was then validated the model in the validation, test, and control sets. It was demonstrated that low weighted predictive scores (wp-scores) were significantly associated with longer PFS compared to high wp-scores in the validation and test sets. It was concluded that the cfDNA 5hmC signature is a robust biomarker for predicting ICI treatment response in lung cancer.

It seemed that all blood samples used for the analysis were collected from lung cancer patients who were on ICI treatment. Although the result showed that the 5hmC predictive signature was significantly correlated with PFS, but independent of age, sex, race, and PD-L1 TPS, there were no data showing that the 5hmC signature was specifically correlated with the PFS in patients receiving ICI. A recent study by Liang, G. et al (Adv Sci (Weinh). 2023 Aug; 10(23): 2300445) revealed the potential of 5hmC gene signature in predicting outcomes of hypomethylating agent‐based therapies in patients with AML, implicating that 5-hmC signatures in cell-free DNA could be a potential biomarkers for lung cancer prognosis and survival independent of the type of treatment. Unless another group of patients who were not on ICI were included for the 5hmC analysis as the comparison, the data of this study could not support the conclusion that cfDNA 5hmC signature is a robust biomarker for predicting ICI treatment response in lung cancer. 

Author Response

In this study, a 5hmC predictive model was developed to quantify the 5hmC levels in 85 plasma cell-free DNA (cfDNA) samples from lung cancer patients receiving ICI monotherapy. The model was then validated the model in the validation, test, and control sets. It was demonstrated that low weighted predictive scores (wp-scores) were significantly associated with longer PFS compared to high wp-scores in the validation and test sets. It was concluded that the cfDNA 5hmC signature is a robust biomarker for predicting ICI treatment response in lung cancer.

Response: We greatly appreciate the reviewer’s positive appraisal of our manuscript. We included additional data based on the reviewer’s constructive comments.

Reviewer’s comments:

  1. It seemed that all blood samples used for the analysis were collected from lung cancer patients who were on ICI treatment. Although the result showed that the 5hmC predictive signature was significantly correlated with PFS, but independent of age, sex, race, and PD-L1 TPS, there were no data showing that the 5hmC signature was specifically correlated with the PFS in patients receiving ICI.

Response: We have included the data for the 5hmC signature (wp-scores) and PFS in patients receiving ICI therapy in Table S3. This analysis directly addresses the correlation between the 5hmC signature and PFS in the context of ICI treatment. Please see line 190 and Table S3.

  1. A recent study by Liang, G. et al (Adv Sci (Weinh). 2023 Aug; 10(23): 2300445) revealed the potential of 5hmC gene signature in predicting outcomes of hypomethylating agent‐based therapies in patients with AML, implicating that 5-hmC signatures in cell-free DNA could be a potential biomarkers for lung cancer prognosis and survival independent of the type of treatment. Unless another group of patients who were not on ICI were included for the 5hmC analysis as the comparison, the data of this study could not support the conclusion that cfDNA 5hmC signature is a robust biomarker for predicting ICI treatment response in lung cancer.

Response: We agree that the predictive value of the cfDNA 5hmC signature should be validated in lung cancer patients who were not on ICIs. We have performed our analysis to include a cohort of 27 late-stage lung cancer patients who did not receive ICI treatment. Our analysis revealed that the wp-scores were not significantly associated with either PFS (P =.66; HR 0.81; 95% CI, 0.32–2.00; Figure S4A) or OS (P =.74; HR 0.83; 95% CI, 0.28–2.50; Figure S4B) in these patients. This finding highlights the specificity of the 5hmC signature in predicting responses to ICI treatments in lung cancer, underscoring its potential as a biomarker specific to this therapeutic context. Please see lines 296-305, Figure S4A, B.

Reviewer 2 Report

Comments and Suggestions for Authors

In this interesting manuscript authors provides proof-of-concept evidence that plasma cfDNA 5hmC can accurately predict ICI treatment response in lung cancer. The manuscript is well written and the results interesting and well presented. I only have 2 minor concerns:

1) Authors should include statistical analyses in all figures

2) Authors should improve the quality of all figures

Author Response

In this interesting manuscript authors provides proof-of-concept evidence that plasma cfDNA 5hmC can accurately predict ICI treatment response in lung cancer. The manuscript is well written and the results interesting and well presented. I only have 2 minor concerns:

Response: We thank the Reviewer for the constructive comments. Based on the reviewer’s suggestions, we have revised the manuscript.

Reviewer’s comments: 

  1. Authors should include statistical analyses in all figures

Response: We have included statistical analyses in all figures. Please see revised figure legends.

  1. Authors should improve the quality of all figures

Response: We have improved the quality of all figures and the resolution is 300 pixels/inch.

Round 2

Reviewer 1 Report

Comments and Suggestions for Authors

The authors have addressed my concerns.